# Evaluating System 1 vs. 2 Reasoning Approaches for Zero-Shot Time Series Forecasting: A Benchmark and Insights

## Abstract

Reasoning ability is crucial for solving challenging tasks. With the advancement of foundation models, such as the emergence of large language models (LLMs), a wide range of reasoning strategies has been proposed, including test-time enhancements, such as Chain-of-Thought, and post-training optimizations, as used in DeepSeek-R1. While these reasoning strategies have demonstrated effectiveness across various challenging language or vision tasks, their applicability and impact on time-series forecasting (TSF), particularly the challenging zero-shot TSF, remain largely unexplored. In particular, it is unclear whether zero-shot TSF benefits from reasoning and, if so, what types of reasoning strategies are most effective.

To bridge this gap, we propose REC4TS, the first benchmark that systematically evaluates the effectiveness of popular reasoning strategies when applied to zero-shot TSF tasks. REC4TS conducts comprehensive evaluations across datasets spanning eight domains, covering unimodal and multimodal settings under short- and long-term forecasting horizons. More importantly, REC4TS provides key insights: (1) Self-consistency emerges as the most effective test-time reasoning strategy; (2) Hybrid reasoning with low-to-moderate reasoning effort often gives the best win-rate trade-off; (3) DeepSeek-R1 performs best among evaluated System 2 models, suggesting that outcome-oriented GRPO-style training may be promising while remaining confounded by model-family and training-data differences; (4) Multimodal TSF benefits more from reasoning strategies compared to unimodal TSF. Beyond these insights, REC4TS establishes two

building blocks to support future zero-shot TSF reasoning research: (1) A novel dataset, TIME-THINKING, containing forecasting samples annotated with reasoning trajectories from multiple advanced LLMs, and (2) A simple test-time scaling trend for foundation TSF models inspired by self-consistency. All data and code are accessible at: https://anonymous.4open.science/r/REC4TS-2700

## 1. Introduction

Reasoning capabilities are fundamental to solving challenging tasks and have been the focus of research for decades (Gebhardt et al., 1997; Wüstenberg et al., 2012). Recent advancements in foundation models, particularly the rise of large language models (LLMs), have brought reasoning strategy research into the era of artificial intelligence (Wei et al., 2022; Wang et al., 2022; Pan et al., 2023). By integrating designed reasoning strategies with powerful foundation models, these models have demonstrated remarkable success in tackling complex real-world challenges, including math, planning, and engineering (Bai et al., 2023; Song et al., 2023; Sun et al., 2023). Despite these advancements in extensive domains, the application of reasoning strategies in zero-shot time-series forecasting (TSF) remains largely unexplored. While recent TSF foundation models have initially demonstrated the feasibility of zero-shot TSF (Das et al., 2024; Goswami et al., 2024; Ansari et al., 2024; Shi et al., 2024), they primarily rely on large-scale TS data from the pre-training phase rather than employing explicit reasoning for zero-shot TSF.

However, reasoning is not only natural but also essential for zero-shot TSF. On the one hand, time-series data inherently contains rich causal relationships, making it a natural domain for reasoning-based approaches (Moraffah et al., 2021; Runge et al., 2023). For instance, in epidemiological forecasting, the outbreak of a pandemic can naturally lead to a subsequent increase in infections—an essential target for flu prediction (Mathis et al., 2024). On the other hand, and more critically, reasoning plays an important role in zero-shot TSF. Unlike conventional supervised TSF, which "memorizes" underlying correlation structures from training data, zero-shot TSF—designed as the primary objective for

---

[1]Anonymous Institution, Anonymous City, Anonymous Region, Anonymous Country. Correspondence to: Anonymous Author <anon.email@domain.com>.

Preliminary work. Under review by the International Conference on Machine Learning (ICML). Do not distribute.

most time-series foundation models—depends on reasoning capabilities to "interpret" the pattern correlations within limited historical data, which have never been "memorized," in order to infer future values. Nevertheless, existing time-series foundation models have yet to fully exploit them by incorporating effective reasoning strategies.

Therefore, a natural twofold research question (RQ) for reasoning strategies on zero-shot TSF arises: **"RQ1: Can zero-shot TSF benefit from enhanced reasoning ability?"** While reasoning is an integral part of time-series analysis, such as used in feature selection and statistical analysis (Sun et al., 2015; Chen et al., 2004), its effectiveness has yet to be systematically validated in zero-shot TSF with foundation models. Furthermore, if reasoning is indeed beneficial (as demonstrated later in our work), a subsequent question emerges: **"RQ2: What kind of reasoning strategies does zero-shot TSF need?"**

Existing reasoning strategies can be organized into four practical approaches: direct System 1, reasoning-enhanced System 1, System 2, and hybrid reasoning. These approaches differ distinctly in their implementation. Reasoning-enhanced System 1 typically occurs during the inference stage, such as Chain-of-Thought (CoT) reasoning, possibly being further enhanced through self-consistency and self-correction (Wei et al., 2022; Wang et al., 2022; Pan et al., 2023; Kumar et al., 2024). In contrast, System 2 reasoning involves deep and analytical reasoning (Kahneman, 2011) built into the foundation model by incorporating incentives and rewards during post-training (Ouyang et al., 2022; Guo et al., 2025). Furthermore, hybrid reasoning adjusts the mixture of System 1 and System 2 by setting the reasoning-effort hyperparameter. Given these fundamental differences, addressing **RQ2** requires an in-depth investigation into their effectiveness for time-series forecasting. However, such studies remain unexplored, and a systematic benchmark to assess the impact of different reasoning strategies in TSF is still lacking.

To bridge this gap and address the proposed twofold research questions, we introduce REC4TS, the first benchmark that systematically evaluates popular reasoning strategies in zero-shot TSF. Our contributions are fourfold

**(a) New research direction:** we study when, what, and how reasoning strategies impact zero-shot TSF.

**(b) Comprehensive benchmark:** we conduct about 2,500 experiments covering reasoning-enhanced System 1, System 2, and hybrid reasoning strategies across nine foundation models.

**(c) In-depth insights:** we find that self-consistency is the most effective reasoning-enhanced System 1 strategy, DeepSeek-R1 performs best among evaluated System 2 models, low-to-moderate hybrid reasoning effort often gives

the best win-rate trade-off, and multimodal zero-shot TSF benefits more from reasoning than unimodal TSF.

**(d) Open-source toolkits:** we release REC4TS, a test-time scaling-trend analysis, and TIME-THINKING, the first TSF dataset with reasoning trajectories.

Related work, detailed experimental results, limitations and future work, and conclusion are provided in App. A, App. B, App. C, and App. D, respectively.

## 2. REC4TS: A Suite of Evaluating Reasoning Strategies for Zero-Shot TSF

REC4TS consists of four core modules: Datasets, Reasoning Strategies, Models, and Evaluations. We introduce these modules one by one in this section. More details are provided in Appendix E.

### 2.1. Dataset module

The dataset module includes datasets from eight different domains, all containing both numerical time series and aligned textual context series, providing unified data support for downstream time series forecasting. As detailed in Table 7, these verified datasets (Liu et al., 2025; Lin et al., 2024a) cover key domains such as Agriculture, Climate, Economy, Energy, Health, Security, Employment, and Traffic, with weekly and monthly frequencies. The textual context series consists of keyword-based web summaries, aligned by date with the numerical series. As shown in Figure 5, these datasets exhibit diverse characteristics, enabling comprehensive evaluation.

### 2.2. Reasoning Strategies Module

As shown in Figure 1, REC4TS includes four reasoning approaches (Pan et al., 2023; Plaat et al., 2024; Xu et al., 2025): (1) Direct System 1 Prediction, which directly uses generative models such as GPT-4o; (2) Test-Time-Enhanced System 1 Reasoning, including Chain-of-Thought (Wei et al., 2022), Self-Consistency (Wang et al., 2022), and Self-Correction (Madaan et al., 2023), which add inference-time computation without updating weights; (3) Post-Training-Enabled System 2 Reasoning, such as DeepSeek-R1 (Guo et al., 2025), which builds reasoning through reinforcement learning and reasoning data; and (4) Hybrid reasoning, which explicitly adjusts reasoning effort to mix fast responses with deeper analytical thinking.

Specifically, CoT decomposes forecasting into steps; Self-Consistency samples multiple forecasts and aggregates them using the median; and Self-Correction refines outputs through feedback. For System 2, o1-mini, Gemini-2.0-flash-thinking, and DeepSeek-R1 are included. Compared with o1-mini's PPO (Schulman et al., 2017) policy-and-critic training, DeepSeek-R1 adopts GRPO (Shao et al., 2024), removing the critic by ranking multiple responses.

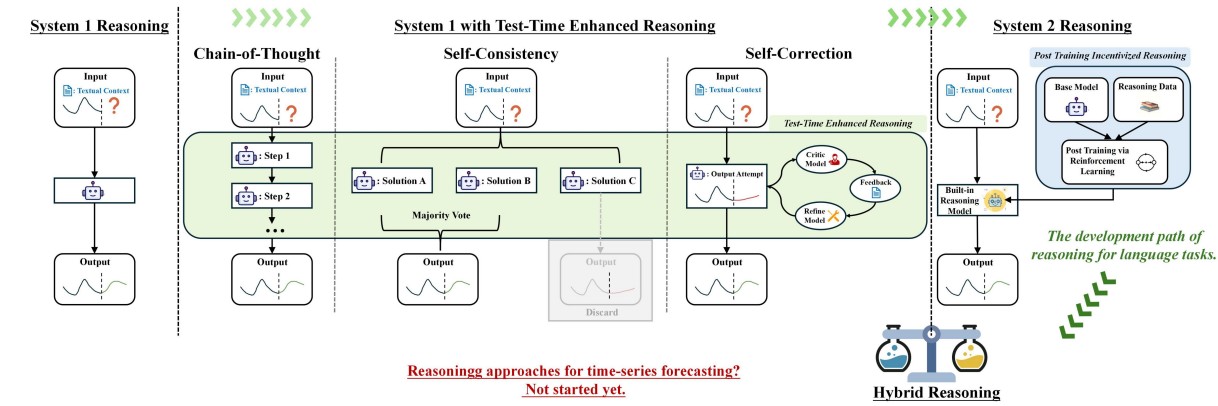

*Figure 1.* The reasoning strategies included in the proposed REC4TS benchmark. REC4TS includes four approaches: direct System 1, test-time-enhanced System 1 with Chain-of-Thought, Self-Consistency, and Self-Correction, post-training-empowered System 2 through reinforcement learning, such as DeepSeek-R1 (Guo et al., 2025), and hybrid reasoning models with adjustable reasoning effort.

## 2.3. Models Module

REC4TS covers three foundational-model series, including closed- and open-source models. Each series provides System 1 and System 2 versions. Since reasoning strategies for foundational time-series models remain underexplored and difficult to implement directly, reusing foundational language models for zero-shot TSF is currently the most practical choice (Xue & Salim, 2023; Gruver et al., 2023; Liu et al., 2024b; Jin et al.; Cao et al.).

Specifically, REC4TS includes OpenAI's GPT-4o and o1-mini, Google's Gemini-2.0-Flash and Gemini-2.0-Flash-Thinking, and DeepSeek's DeepSeek-V3 and DeepSeek-R1 as the corresponding System 1 and System 2 pairs, respectively. Inspired by recent research (Wang et al., 2024a; Hoo et al., 2025; Hu et al., 2025), we reformulate numerical time series into a tabular format, i.e., "timestamp : numerical value", to enable LLMs as powerful time-series analysts.

Furthermore, REC4TS includes hybrid reasoning models beyond pure System 1 and System 2: OpenAI's o4-mini, Google's Gemini-2.5-Pro, and Anthropic's Claude-3.7-Sonnet. We evaluate low, medium, and high reasoning effort; o4-mini exposes these levels directly, while Gemini-2.5-Pro and Claude-3.7-Sonnet are approximated with 1024, 2048, and 4096 reasoning tokens. GPT-4o is the reference model for the hybrid comparison, with detailed budget-level results in Appendix (Tables 4, 5, and 6).

## 2.4. Evaluation Module

We consider four settings: unimodal short-term, unimodal long-term, multi-modal short-term, and multi-modal long-term. Multi-modal TSF uses numerical series plus aligned textual context, whereas unimodal TSF uses only numerical series. Long-term TSF forecasts the next six months and short-term TSF forecasts the next three months. Following common TSF practice (Wu et al., 2021; 2022; Nie et al.), we set the lookback window to 96, use Mean Squared Error

(MSE), and avoid data contamination by evaluating horizon windows after October 2023, the knowledge cutoff of selected foundation models.

## 3. Experimental Results and Insights

Detailed experimental results are provided in App. B, including full OpenAI, Gemini, DeepSeek, and hybrid-reasoning budget tables across the four settings. Based on the constructed REC4TS suite, we evaluate zero-shot TSF reasoning strategies across eight datasets and four settings, repeating each experiment three times and reporting mean/std MSE. Figure 3 visualizes win rate over direct System 1, where 50% indicates no aggregate advantage over direct System 1, and Figure 2 summarizes hybrid reasoning under different effort levels. We discuss the two RQs from Section 1 using Answer–Evidence–Analysis.

### 3.1. RQ1: Can TSF Benefit from Reasoning?

**Answer.** TSF benefits from enhanced reasoning: in all four scenarios, at least two strategies outperform their System 1 counterparts in over 50% of cases, and at least one exceeds 60%. **Evidence and analysis.** Across the full tables, long-term TSF benefits more consistently: CoT, Self-Consistency, and Self-Correction win in 52.08%, 58.33%, and 54.17% of long-term cases. Multimodal TSF benefits more significantly: Self-Consistency and Self-Correction win in 66.67% and 56.25% of multimodal cases. Thus, long horizons and textual context require more explicit reasoning over temporal and contextual factors.

### 3.2. RQ2: What Reasoning Strategies Does TSF Need?

**Answer.** Self-consistency is the most reliable plug-in strategy, outperforming System 1 in 54.17%–79.17% of comparisons across settings; hybrid reasoning is also effective when effort stays low, suggesting TSF needs fast pattern recognition plus controlled deeper reasoning. **Evidence and analysis.** Test-time reasoning strategies, especially

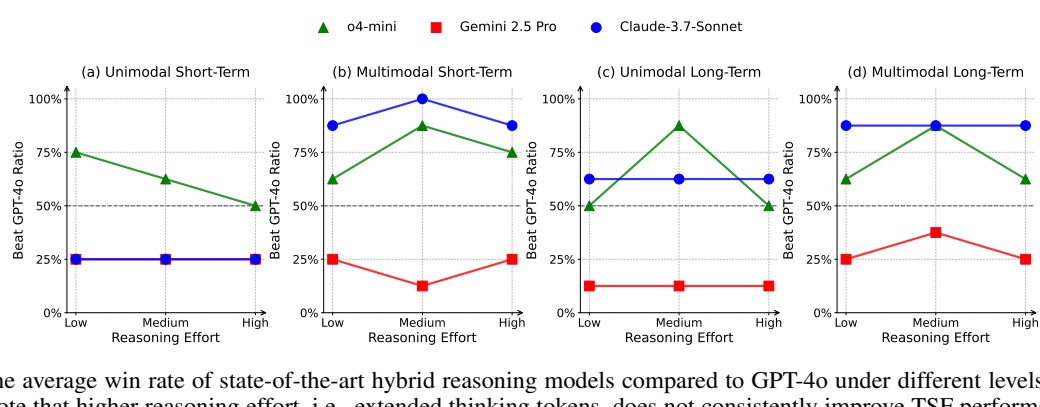

*Figure 2.* The average win rate of state-of-the-art hybrid reasoning models compared to GPT-4o under different levels of reasoning effort. We note that higher reasoning effort, i.e., extended thinking tokens, does not consistently improve TSF performance. Overall, low-to-moderate effort often gives the best win-rate trade-off. Detailed budget-level results are provided in App. Tables 4, 5, and 6.

self-consistency, are more consistently effective than pure System 2 models across the evaluated settings. Among System 2 models, DeepSeek-R1 is the only effective one, improving in three of four settings and improving over DeepSeek-V3 in 18 out of 32 comparisons (56.25%), while o1-mini and Gemini-thinking mostly fail. We hypothesize that DeepSeek-R1's outcome-oriented GRPO-style training (Guo et al., 2025) may better match TSF, where exact reasoning traces are hard to label; however, this comparison is confounded by model-family and training-data differences. For hybrid reasoning, the budget-level results in App. Tables 4, 5, and 6 show that o4-mini and Claude-3.7-Sonnet are strongest under low or moderate effort, and low-to-moderate effort often gives the best win-rate trade-off. Overall, TSF benefits more from diverse test-time paths and calibrated effort than generic long CoT.

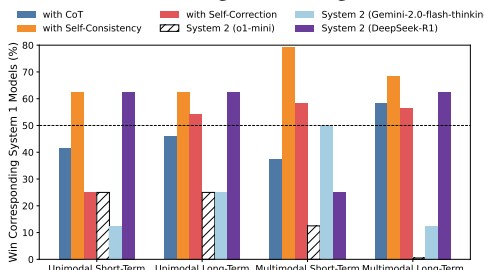

*Figure 3.* The average win rate of reasoning strategies compared to corresponding direct System 1. We observe the consistent and significant effectiveness of self-consistency, as well as the unique effectiveness of DeepSeek-R1 among System 2 strategies.

## 4. Open-Source Toolkits

We release three resources: RᴇC4TS, an evaluation suite; a test-time scaling trend; and Tɪᴍᴇ-Tʜɪɴᴋɪɴɢ, the first TSF dataset with reasoning trajectories.

**Evaluation suite.** RᴇC4TS supports batch experiments, unified LLM/reasoning-strategy selection, automated result extraction, and comparison across uni-/multimodal and short-/long-term settings. It includes hyperparameters and outputs for about 2,500 pure/hybrid reasoning experiments.

**Observed Scaling Trend.** To transfer self-consistency effectiveness to foundation TSF models, we treat the sampling

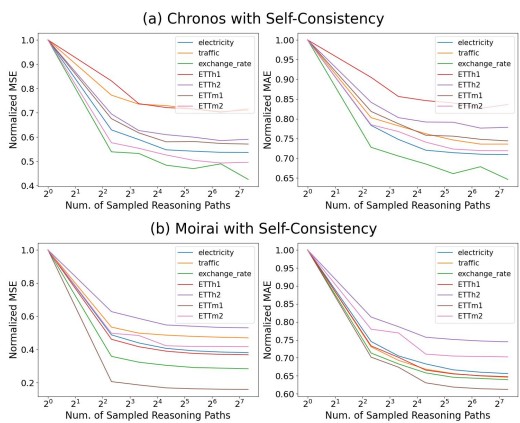

*Figure 4.* Observed test-time scaling trend on foundation time-series models (Chronos and Moirai) inspired by our insights. MSE and MAE are normalized based on one sampled path's performances. The performances continuously improve as the number of sampled reasoning paths in the self-consistency strategy increases.

number of probabilistic models, including Moirai (Woo et al., 2024) and Chronos (Ansari et al., 2024), as reasoning paths and use the median as the consistent prediction. As shown in Figure 4, both models improve as paths increase, reducing MSE by 20%–50% and converging around 32 paths; more settings are in App. F.

**Reasoning-annotated TSF Dataset.** Inspired by recent research in post-training large reasoning models (Zhou et al., 2023; Wang et al., 2024b; Muennighoff et al., 2025), we realized that another obstacle to research on foundation TSF reasoning models, aside from evaluation suites and scaling trends, is the lack of a reasoning-annotated dataset. To this end, we curate the first reasoning-annotated TSF dataset, named Tɪᴍᴇ-Tʜɪɴᴋɪɴɢ, which pairs TSF task queries and answers with reasoning traces. Specifically, we adopt six advanced and diverse foundation models, including GPT-4o, o1-mini, Gemini-2.0-flash, Gemini-2.0-flash-thinking, DeepSeek-V3, and DeepSeek-R1, and record both the visible final output and the intermediate reasoning chain (if available) for TSF tasks. We repeat the sampling 15 times, covering all datasets and settings in RᴇC4TS, and select 2K high-quality samples. See App. G for more details.

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

## A. Related Work

**(Zero-Shot) Time-Series Forecasting.** Time-series forecasting (TSF) is a fundamental task in time-series analysis with broad real-world applications, including economics, urban computing, and epidemiology (Sezer et al., 2020; Tabassum et al., 2021; Rodríguez et al., 2024). The introduction of transformer-based architectures (Vaswani et al., 2017) has significantly advanced TSF performance, setting new benchmarks in forecasting accuracy (Zhou et al., 2021; 2022; Nie et al., 2023; Liu et al., 2023). Building upon these advancements, recent research has explored foundation TSF models, which achieve competitive zero-shot forecasting performance comparable to supervised TSF models (Das et al., 2024; Ansari et al., 2024; Goswami et al., 2024; Shi et al., 2024; Ekambaram et al., 2025; Kamarthi & Prakash, 2023). Trained on billions of data samples, these foundation models can generate accurate forecasts across diverse time series without requiring additional post-training or fine-tuning.

**Foundation Model Reasoning.** With the rise of foundation models, reasoning has become a critical research direction, enabling models to move beyond surface-level pattern recognition toward structured decision-making (Wei et al., 2022; Brown et al., 2020). Reasoning in foundation models primarily follows two paradigms: reasoning-enhanced System 1, which improves inference during test time for fast and intuitive thinking, such as Chain-of-Thought reasoning (Wei et al., 2022; Wang et al., 2022; Pan et al., 2023; Kumar et al., 2024), and System 2, which enables deep, analytical reasoning as an inherent capability of the model. System 2 reasoning is typically developed through post-training techniques such as reinforcement learning from human feedback (RLHF), reward shaping, knowledge distillation, and the more recent group-relative policy optimization (GRPO) in DeepSeek-R1 (Bai et al., 2022; Kwon et al., 2023; Gou et al., 2021; Xu et al., 2024; Guo et al., 2025), which allows foundation models to process more structured, deeper,

and analytical thinking. Leveraging these advanced reasoning strategies, foundation models have demonstrated strong capabilities in handling complex tasks, including mathematical problem-solving, automated planning, and applications in engineering and science (Bai et al., 2023; Zhou et al., 2024; Sun et al., 2023; Song et al., 2023; Guo et al., 2023; Goh et al., 2024).

**Time-Series Reasoning.** Reasoning over time-series data is an important topic in time-series analysis (Kauppinen & Hyvönen, 2007). Conventional time-series reasoning primarily focuses on causal analysis, including causal discovery, feature selection, and graph-based analysis (Eichler & Didelez, 2012; Sun et al., 2015; Runge et al., 2023; Chen et al., 2004). However, these conventional methods focus only on numerical modalities, making validation challenging and often limiting their application to synthetic or simplified scenarios. Recent advancements have broadened time-series reasoning beyond numerical data, incorporating multimodal perspectives. Notably, emerging research has explored reasoning over time-series data using LLMs (Liu et al., 2024b; Yan et al., 2025) and vision-based approaches (Liu et al., 2024a), expanding its scope and applicability. While these studies show promising potential for time series reasoning with advanced foundation models, they do not explicitly establish a connection between reasoning strategies and TSF tasks.

Importantly, existing studies and benchmarks focus on either (zero-shot) TSF (Aksu et al., 2024; Wang et al., 2024b; Du et al., 2024) or reasoning strategies (Lin et al., 2024b; Hao et al., 2024; Zhang et al., 2025), no existing work systematically evaluates how different reasoning strategies impact zero-shot TSF. Our study bridges this gap and introduces a multimodal reasoning enhancement setup beyond conventional unimodal approaches. This multimodal setup incorporates textual context alongside numerical data, providing more comprehensive information for real-world forecasting, and setting a higher standardization for reasoning ability in zero-shot TSF tasks.

## B. Detailed Experimental Results

The full result tables are grouped by model family and hybrid-reasoning budget below.

## B.1. OpenAI Results

*Table 1.* Results with OpenAI's System 1 (GPT-4o) and 2 (o1-mini) Models. We report the mean MSE and standard deviation over three repeated experiments. Reasoning strategies that outperform the direct System 1 are highlighted in green, while those that perform worse or have similar performance are marked in red. In "Win System 1," we present the probability of each reasoning strategy outperforming System 1 across datasets. We observe that only the self-consistency strategy is consistently effective, while the System 2 strategy is consistently ineffective.

*(a)* Results of Unimodal Short-term TSF Settings. We use numerical series only to forecast the next three months.

| Dataset | System 1 | System 1 with Test-time Reasoning Enhancement | | | System 2 |
|---|---|---|---|---|---|
| | GPT-4o | with CoT | with Self-Consistency | with Self-Correction | o1-mini |
| Agriculture | 0.021±0.011 | 0.909±1.275 | 0.021±0.003 | 0.025±0.007 | 0.069±0.013 |
| Climate | 1.599±0.500 | 1.704±0.164 | 1.517±0.263 | 1.998±0.677 | 1.412±0.159 |
| Economy | 0.631±0.135 | 0.638±0.410 | 0.450±0.171 | 1.018±0.184 | 0.583±0.001 |
| Energy | 0.363±0.110 | 0.258±0.029 | 0.167±0.242 | 0.396±0.086 | 0.930±0.747 |
| Flu | 0.568±0.425 | 0.592±0.291 | 0.481±0.288 | 0.663±0.078 | 1.441±1.234 |
| Security | 0.093±0.029 | 0.259±0.001 | 0.084±0.028 | 0.165±0.070 | 0.225±0.048 |
| Employment | 0.010±0.004 | 0.006±0.002 | 0.012±0.001 | 0.013±0.003 | 0.021±0.003 |
| Traffic | 0.385±0.471 | 0.113±0.063 | 0.047±0.009 | 0.053±0.009 | 0.566±0.731 |
| Win System 1 | NA | 3/8 | 5/8 | 1/8 | 2/8 |

*(b)* Results of Multimodal Short-term TSF Settings. We use numerical series with textual context series to forecast the next three months.

| Dataset | System 1 | System 1 with Test-time Reasoning Enhancement | | | System 2 |
|---|---|---|---|---|---|
| | GPT-4o | with CoT | with Self-Consistency | with Self-Correction | o1-mini |
| Agriculture | 0.018±0.015 | 0.018±0.011 | 0.013±0.008 | 0.018±0.006 | 0.045±0.056 |
| Climate | 1.716±0.580 | 1.920±0.505 | 1.712±0.191 | 2.042±0.609 | 1.603±0.496 |
| Economy | 0.569±0.162 | 0.940±0.445 | 0.291±0.127 | 0.503±0.071 | 0.583±0.001 |
| Energy | 0.541±0.457 | 0.316±0.125 | 0.187±0.090 | 0.225±0.080 | 0.718±0.786 |
| Flu | 0.548±0.164 | 1.071±0.643 | 0.288±0.071 | 1.261±1.164 | 0.983±1.177 |
| Security | 0.076±0.052 | 0.110±0.087 | 0.146±0.025 | 0.151±0.035 | 0.244±0.020 |
| Employment | 0.020±0.006 | 0.020±0.003 | 0.019±0.003 | 0.021±0.004 | 0.028±0.008 |
| Traffic | 0.551±0.396 | 1.577±1.421 | 0.030±0.010 | 0.347±0.349 | 0.911±0.594 |
| Win System 1 | NA | 2/8 | 7/8 | 4/8 | 1/8 |

*(c)* Results of Unimodal Long-term TSF Settings. We use numerical series only to forecast the next six months.

| Dataset | System 1 | System 1 with Test-time Reasoning Enhancement | | | System 2 |
|---|---|---|---|---|---|
| | GPT-4o | with CoT | with Self-Consistency | with Self-Correction | o1-mini |
| Agriculture | 0.093±0.057 | 0.920±1.134 | 0.057±0.011 | 0.068±0.018 | 0.293±0.089 |
| Climate | 0.754±0.051 | 1.199±0.132 | 0.811±0.081 | 0.877±0.041 | 0.708±0.058 |
| Economy | 0.463±0.146 | 1.040±0.482 | 0.620±0.116 | 0.748±0.069 | 0.359±0.001 |
| Energy | 0.197±0.038 | 0.746±0.500 | 0.177±0.062 | 0.296±0.153 | 0.926±0.771 |
| Flu | 0.219±0.053 | 0.967±0.412 | 0.230±0.077 | 0.639±0.479 | 0.862±0.597 |
| Security | 0.183±0.044 | 0.162±0.038 | 0.135±0.011 | 0.165±0.017 | 0.211±0.075 |
| Employment | 0.011±0.006 | 0.013±0.002 | 0.009±0.003 | 0.013±0.004 | 0.053±0.015 |
| Traffic | 0.066±0.046 | 0.218±0.158 | 0.046±0.016 | 0.036±0.008 | 0.091±0.042 |
| Win System 1 | NA | 1/8 | 5/8 | 3/8 | 2/8 |

*(d)* Results of Multimodal Long-term TSF Settings. We use numerical series with textual context series to forecast the next six months.

| Dataset | System 1 | System 1 with Test-time Reasoning Enhancement | | | System 2 |
|---|---|---|---|---|---|
| | GPT-4o | with CoT | with Self-Consistency | with Self-Correction | o1-mini |
| Agriculture | 0.110±0.065 | 0.097±0.044 | 0.063±0.009 | 0.051±0.042 | 0.210±0.022 |
| Climate | 1.365±0.479 | 0.995±0.109 | 1.065±0.014 | 0.912±0.004 | 1.549±0.566 |
| Economy | 0.487±0.237 | 1.027±0.321 | 0.500±0.184 | 0.543±0.074 | 0.827±0.662 |
| Energy | 0.365±0.185 | 0.254±0.122 | 33.743±23.911 | 0.293±0.026 | 0.707±0.499 |
| Flu | 0.291±0.065 | 0.369±0.058 | 0.445±0.210 | 0.529±0.365 | 1.070±0.284 |
| Security | 0.196±0.056 | 0.188±0.027 | 0.140±0.028 | 0.116±0.041 | 0.207±0.001 |
| Employment | 0.015±0.002 | 0.021±0.007 | 0.021±0.002 | 0.106±0.115 | 0.031±0.003 |
| Traffic | 0.207±0.205 | 0.341±0.402 | 0.045±0.013 | 0.377±0.504 | 1.482±1.788 |
| Win System 1 | NA | 4/8 | 4/8 | 4/8 | 0/8 |

## B.2. Gemini Results

*Table 2.* Results with Google's System 1 (Gemini-2.0-flash) and 2 (Gemini-2.0-flash-thinking) Models. We report the mean MSE and standard deviation over three repeated experiments. Reasoning strategies that outperform the direct System 1 are highlighted in green, while those that perform worse or have similar performance are marked in red. In "Win System 1," we present the probability of each reasoning strategy outperforming System 1 across datasets. We again observe that only the self-consistency strategy consistently works, while the System 2 strategy consistently fails.

*(a)* Results of Unimodal Short-term TSF Settings. We use numerical series only to forecast the next three months.

| Dataset | System 1 | System 1 with Test-time Reasoning Enhancement | | | System 2 |
|---|---|---|---|---|---|
| | Gemini-2.0-flash | with CoT | with Self-Consistency | with Self-Correction | Gemini-2.0-flash-thinking |
| Agriculture | 0.011±0.001 | 0.010±0.004 | 0.009±0.004 | 0.012±0.008 | 0.017±0.004 |
| Climate | 1.234±0.239 | 1.800±0.326 | 1.749±0.791 | 1.703±0.280 | 2.416±0.112 |
| Economy | 0.113±0.007 | 0.272±0.256 | 0.229±0.145 | 0.121±0.026 | 0.172±0.049 |
| Energy | 0.172±0.038 | 0.181±0.048 | 0.132±0.047 | 0.235±0.060 | 0.327±0.054 |
| Flu | 0.809±0.353 | 0.641±0.224 | 0.402±0.197 | 1.854±1.271 | 2.068±1.076 |
| Security | 0.170±0.054 | 0.252±0.104 | 0.380±0.323 | 0.191±0.095 | 0.259±0.001 |
| Employment | 0.002±0.001 | 0.005±0.003 | 0.004±0.004 | 0.004±0.002 | 0.311±0.001 |
| Traffic | 0.347±0.415 | 0.097±0.060 | 0.016±0.006 | 0.034±0.014 | 0.201±0.001 |
| Win System 1 | NA | 3/8 | 4/8 | 1/8 | 1/8 |

*(b)* Results of Multimodal Short-term TSF Settings. We use numerical series with textual context series to forecast the next three months.

| Dataset | System 1 | System 1 with Test-time Reasoning Enhancement | | | System 2 |
|---|---|---|---|---|---|
| | Gemini-2.0-flash | with CoT | with Self-Consistency | with Self-Correction | Gemini-2.0-flash-thinking |
| Agriculture | 0.010±0.003 | 0.006±0.001 | 0.009±0.002 | 0.011±0.004 | 0.008±0.002 |
| Climate | 2.115±0.660 | 1.725±0.227 | 1.980±0.760 | 1.529±0.290 | 2.106±0.294 |
| Economy | 0.376±0.085 | 0.326±0.067 | 0.373±0.079 | 0.283±0.083 | 0.509±0.109 |
| Energy | 0.143±0.069 | 0.117±0.015 | 0.143±0.027 | 0.091±0.065 | 0.218±0.106 |
| Flu | 0.594±0.219 | 0.607±0.294 | 0.332±0.102 | 1.422±0.542 | 3.171±0.001 |
| Security | 0.558±0.604 | 0.145±0.050 | 0.172±0.119 | 0.141±0.065 | 0.259±0.001 |
| Employment | 0.013±0.002 | 0.015±0.002 | 0.011±0.002 | 0.011±0.003 | 0.311±0.001 |
| Traffic | 0.322±0.196 | 0.046±0.017 | 0.163±0.106 | 0.425±0.235 | 0.201±0.001 |
| Win System 1 | NA | 6/8 | 7/8 | 5/8 | 4/8 |

*(c)* Results of Unimodal Long-term TSF Settings. We use numerical series only to forecast the next six months.

| Dataset | System 1 | System 1 with Test-time Reasoning Enhancement | | | System 2 |
|---|---|---|---|---|---|
| | Gemini-2.0-flash | with CoT | with Self-Consistency | with Self-Correction | Gemini-2.0-flash-thinking |
| Agriculture | 0.032±0.007 | 0.036±0.011 | 0.035±0.007 | 0.077±0.026 | 0.093±0.018 |
| Climate | 1.476±0.651 | 0.964±0.321 | 0.674±0.092 | 0.908±0.153 | 1.240±0.705 |
| Economy | 0.092±0.038 | 0.216±0.142 | 0.078±0.013 | 0.066±0.003 | 0.244±0.035 |
| Energy | 0.303±0.044 | 0.130±0.021 | 0.241±0.060 | 0.489±0.134 | 0.241±0.148 |
| Flu | 1.190±1.171 | 1.049±0.447 | 0.596±0.128 | 1.095±0.546 | 1.920±0.001 |
| Security | 0.196±0.052 | 0.533±0.493 | 0.955±0.389 | 0.154±0.031 | 0.207±0.001 |
| Employment | 0.011±0.001 | 0.019±0.007 | 0.009±0.002 | 0.013±0.005 | 0.268±0.001 |
| Traffic | 0.068±0.063 | 0.215±0.079 | 0.074±0.048 | 0.050±0.013 | 0.414±0.001 |
| Win System 1 | NA | 3/8 | 5/8 | 5/8 | 2/8 |

*(d)* Results of Multimodal Long-term TSF Settings. We use numerical series with textual context series to forecast the next six months.

| Dataset | System 1 | System 1 with Test-time Reasoning Enhancement | | | System 2 |
|---|---|---|---|---|---|
| | Gemini-2.0-flash | with CoT | with Self-Consistency | with Self-Correction | Gemini-2.0-flash-thinking |
| Agriculture | 0.052±0.026 | 0.034±0.009 | 0.034±0.006 | 0.024±0.007 | 0.096±0.032 |
| Climate | 1.644±0.398 | 1.452±0.461 | 1.318±0.079 | 1.292±0.401 | 1.006±0.327 |
| Economy | 0.092±0.010 | 0.234±0.049 | 0.134±0.044 | 10.357±14.475 | 1.093±0.806 |
| Energy | 0.138±0.106 | 0.208±0.116 | 0.159±0.077 | 0.384±0.074 | 0.713±0.513 |
| Flu | 0.659±0.173 | 0.557±0.164 | 0.477±0.006 | 0.785±0.064 | 1.920±0.001 |
| Security | 0.123±0.062 | 0.109±0.020 | 0.142±0.043 | 0.151±0.053 | 0.207±0.001 |
| Employment | 0.029±0.004 | 0.022±0.003 | 0.026±0.003 | 0.026±0.002 | 0.268±0.001 |
| Traffic | 0.085±0.068 | 0.037±0.027 | 0.020±0.007 | 0.058±0.010 | 0.414±0.001 |
| Win System 1 | NA | 6/8 | 5/8 | 4/8 | 1/8 |

## B.3. DeepSeek Results

*Table 3.* Results with DeepSeek's System 1 (DeepSeek-V3) and 2 (DeepSeek-R1) Models. We report the mean MSE and standard deviation over three repeated experiments. Reasoning strategies that outperform the direct System 1 are highlighted in green, while those that perform worse or have similar performance are marked in red. In "Win System 1," we present the probability of each reasoning strategy outperforming System 1 across datasets. We observe that the self-consistency still consistently works. We find that DeepSeek-R1 is the only evaluated System 2 model that is effective for TSF. We hypothesize that its outcome-oriented GRPO-style training may be helpful, while this comparison remains confounded by model-family and training-data differences.

*(a)* Results of Unimodal Short-term TSF Settings. We use numerical series only to forecast the next three months.

| Dataset | System 1 | System 1 with Test-time Reasoning Enhancement | | | System 2 |
|---|---|---|---|---|---|
| | DeepSeek-V3 | with CoT | with Self-Consistency | with Self-Correction | DeepSeek-R1 |
| Agriculture | 0.038±0.032 | 0.019±0.001 | 0.046±0.015 | 0.013±0.003 | 0.016±0.010 |
| Climate | 1.216±0.202 | 2.650±0.905 | 1.207±0.197 | 1.246±0.081 | 1.541±0.397 |
| Economy | 0.406±0.218 | 0.433±0.031 | 0.284±0.227 | 0.441±0.161 | 0.583±0.001 |
| Energy | 0.736±0.752 | 0.212±0.022 | 0.187±0.011 | 0.182±0.063 | 0.189±0.021 |
| Flu | 1.464±1.031 | 1.650±0.236 | 0.980±0.445 | 1.682±0.292 | 1.298±1.330 |
| Security | 0.283±0.140 | 0.218±0.093 | 0.185±0.052 | 0.116±0.012 | 0.247±0.017 |
| Employment | 0.036±0.019 | 0.020±0.006 | 0.035±0.019 | 0.018±0.007 | 0.012±0.005 |
| Traffic | 0.066±0.031 | 0.201±0.001 | 0.109±0.028 | 0.107±0.067 | 0.113±0.073 |
| Win System 1 | NA | 4/8 | 6/8 | 4/8 | 5/8 |

*(b)* Results of Multimodal Short-term TSF Settings. We use numerical series with textual context series to forecast the next three months.

| Dataset | System 1 | System 1 with Test-time Reasoning Enhancement | | | System 2 |
|---|---|---|---|---|---|
| | DeepSeek-V3 | with CoT | with Self-Consistency | with Self-Correction | DeepSeek-R1 |
| Agriculture | 0.032±0.012 | 0.027±0.006 | 0.023±0.001 | 0.042±0.025 | 2.712±0.001 |
| Climate | 1.428±0.432 | 1.857±0.431 | 1.371±0.001 | 1.411±0.258 | 2.235±0.850 |
| Economy | 0.427±0.174 | 0.598±0.069 | 0.306±0.005 | 0.369±0.128 | 0.615±0.101 |
| Energy | 0.253±0.089 | 0.486±0.318 | 0.197±0.001 | 0.505±0.339 | 0.731±0.777 |
| Flu | 1.073±0.447 | 1.564±0.982 | 0.362±0.161 | 0.441±0.173 | 1.329±1.306 |
| Security | 0.186±0.001 | 0.206±0.010 | 0.187±0.001 | 0.130±0.018 | 0.161±0.051 |
| Employment | 0.016±0.001 | 0.022±0.003 | 0.016±0.001 | 0.016±0.001 | 0.114±0.139 |
| Traffic | 0.201±0.001 | 0.201±0.001 | 0.201±0.001 | 0.114±0.063 | 0.153±0.069 |
| Win System 1 | NA | 1/8 | 5/8 | 5/8 | 2/8 |

*(c)* Results of Unimodal Long-term TSF Settings. We use numerical series only to forecast the next six months.

| Dataset | System 1 | System 1 with Test-time Reasoning Enhancement | | | System 2 |
|---|---|---|---|---|---|
| | DeepSeek-V3 | with CoT | with Self-Consistency | with Self-Correction | DeepSeek-R1 |
| Agriculture | 0.216±0.049 | 0.102±0.034 | 0.103±0.014 | 0.121±0.065 | 0.091±0.019 |
| Climate | 0.902±0.001 | 1.383±0.227 | 0.786±0.153 | 0.913±0.078 | 0.662±0.051 |
| Economy | 0.613±0.776 | 0.540±0.386 | 0.393±0.113 | 0.948±0.589 | 0.359±0.001 |
| Energy | 0.603±0.359 | 0.575±0.452 | 0.923±0.265 | 0.332±0.150 | 1.396±0.001 |
| Flu | 0.841±0.215 | 0.658±0.227 | 0.538±0.021 | 0.939±0.328 | 0.972±0.533 |
| Security | 0.275±0.060 | 0.245±0.039 | 0.280±0.004 | 0.186±0.033 | 0.168±0.028 |
| Employment | 0.051±0.013 | 0.021±0.002 | 0.039±0.006 | 0.023±0.003 | 0.021±0.001 |
| Traffic | 0.414±0.001 | 0.209±0.145 | 94.305±66.620 | 0.306±0.153 | 0.158±0.181 |
| Win System 1 | NA | 7/8 | 5/8 | 5/8 | 6/8 |

*(d)* Results of Multimodal Long-term TSF Settings. We use numerical series with textual context series to forecast the next six months.

| Dataset | System 1 | System 1 with Test-time Reasoning Enhancement | | | System 2 |
|---|---|---|---|---|---|
| | DeepSeek-V3 | with CoT | with Self-Consistency | with Self-Correction | DeepSeek-R1 |
| Agriculture | 0.088±0.058 | 0.063±0.022 | 0.136±0.080 | 0.119±0.078 | 0.019±0.010 |
| Climate | 0.897±0.001 | 2.193±0.330 | 0.897±0.001 | 0.939±0.074 | 1.849±0.570 |
| Economy | 0.629±0.147 | 0.558±0.282 | 0.486±0.074 | 0.623±0.218 | 0.806±0.354 |
| Energy | 0.995±0.139 | 1.286±0.568 | 0.809±0.241 | 0.493±0.112 | 0.746±0.459 |
| Flu | 2.624±2.400 | 0.974±0.446 | 0.644±0.488 | 1.135±0.643 | 1.560±0.957 |
| Security | 0.179±0.002 | 0.250±0.024 | 0.156±0.027 | 0.274±0.071 | 0.134±0.055 |
| Employment | 0.034±0.001 | 0.029±0.008 | 0.034±0.001 | 0.030±0.005 | 0.105±0.115 |
| Traffic | 0.414±0.001 | 0.414±0.001 | 0.414±0.001 | 0.192±0.157 | 0.152±0.185 |
| Win System 1 | NA | 4/8 | 4/8 | 5/8 | 5/8 |

## B.4. Hybrid Reasoning Results

*Table 4.* Performance Comparison of Different Reasoning budgets for o4-mini. We report mean MSE and standard deviation across datasets. Results that outperform GPT-4o are highlighted in green, while those performing worse are marked in red. In "Beat GPT-4o" row, we present the fraction of datasets where each reasoning budget outperforms GPT-4o.

*(a)* Results of Unimodal Short-term TSF Settings. We use numerical series only to forecast the next three months.

| Dataset | GPT-4o | o4-mini low reasoning effort | o4-mini medium reasoning effort | o4-mini high reasoning effort |
|---|---|---|---|---|
| Agriculture | 0.021±0.011 | 0.021±0.013 | 0.044±0.013 | 0.026±0.006 |
| Climate | 1.599±0.500 | 0.960±0.081 | 1.448±0.058 | 1.643±0.109 |
| Economy | 0.631±0.135 | 0.271±0.074 | 0.182±0.065 | 0.165±0.061 |
| Energy | 0.363±0.110 | 0.262±0.131 | 0.168±0.105 | 0.091±0.014 |
| Flu | 0.568±0.425 | 0.508±0.246 | 1.843±1.143 | 2.011±0.987 |
| Security | 0.093±0.029 | 0.216±0.061 | 0.218±0.032 | 0.146±0.022 |
| Employment | 0.010±0.004 | 0.012±0.005 | 0.009±0.005 | 0.005±0.003 |
| Traffic | 0.385±0.471 | 0.069±0.037 | 0.025±0.008 | 0.024±0.010 |
| Beat GPT-4o | NA | 6/8 | 5/8 | 4/8 |

*(b)* Results of Multimodal Short-term TSF Settings. We use numerical series with textual context series to forecast the next three months.

| Dataset | GPT-4o | o4-mini low reasoning effort | o4-mini medium reasoning effort | o4-mini high reasoning effort |
|---|---|---|---|---|
| Agriculture | 0.018±0.015 | 0.018±0.009 | 0.014±0.008 | 0.026±0.022 |
| Climate | 1.716±0.580 | 1.026±0.397 | 1.434±0.529 | 1.660±0.179 |
| Economy | 0.569±0.162 | 0.430±0.217 | 0.212±0.118 | 0.183±0.087 |
| Energy | 0.541±0.457 | 0.181±0.086 | 0.305±0.213 | 0.241±0.247 |
| Flu | 0.548±0.164 | 1.053±0.156 | 0.951±0.175 | 0.371±0.142 |
| Security | 0.076±0.052 | 0.109±0.024 | 0.065±0.032 | 0.143±0.027 |
| Employment | 0.020±0.006 | 0.015±0.005 | 0.016±0.000 | 0.008±0.005 |
| Traffic | 0.551±0.396 | 0.168±0.047 | 0.195±0.221 | 0.023±0.011 |
| Beat GPT-4o | NA | 5/8 | 7/8 | 6/8 |

*(c)* Results of Unimodal Long-term TSF Settings. We use numerical series only to forecast the next six months.

| Dataset | GPT-4o | o4-mini low reasoning effort | o4-mini medium reasoning effort | o4-mini high reasoning effort |
|---|---|---|---|---|
| Agriculture | 0.093±0.057 | 0.026±0.014 | 0.035±0.007 | 0.045±0.029 |
| Climate | 0.754±0.051 | 0.477±0.077 | 0.722±0.103 | 0.694±0.057 |
| Economy | 0.463±0.146 | 0.284±0.053 | 0.106±0.059 | 0.272±0.071 |
| Energy | 0.197±0.038 | 0.392±0.151 | 0.167±0.089 | 0.259±0.194 |
| Flu | 0.219±0.053 | 0.696±0.317 | 1.657±1.006 | 2.379±0.999 |
| Security | 0.183±0.044 | 0.151±0.043 | 0.147±0.076 | 0.185±0.125 |
| Employment | 0.011±0.006 | 0.018±0.004 | 0.009±0.003 | 0.021±0.003 |
| Traffic | 0.066±0.046 | 0.209±0.162 | 0.045±0.009 | 0.038±0.009 |
| Beat GPT-4o | NA | 4/8 | 7/8 | 4/8 |

*(d)* Results of Multimodal Long-term TSF Settings. We use numerical series with textual context series to forecast the next six months.

| Dataset | GPT-4o | o4-mini low reasoning effort | o4-mini medium reasoning effort | o4-mini high reasoning effort |
|---|---|---|---|---|
| Agriculture | 0.110±0.065 | 0.023±0.011 | 0.034±0.004 | 0.875±1.165 |
| Climate | 1.365±0.479 | 0.608±0.058 | 0.988±0.522 | 0.850±0.128 |
| Economy | 0.487±0.237 | 0.264±0.089 | 0.201±0.124 | 0.283±0.176 |
| Energy | 0.365±0.185 | 0.492±0.104 | 0.168±0.092 | 0.088±0.035 |
| Flu | 0.291±0.065 | 0.625±0.290 | 0.916±0.153 | 0.712±0.413 |
| Security | 0.196±0.056 | 0.187±0.029 | 0.164±0.042 | 0.175±0.075 |
| Employment | 0.015±0.002 | 0.021±0.003 | 0.013±0.007 | 0.021±0.012 |
| Traffic | 0.207±0.205 | 0.057±0.048 | 0.130±0.138 | 0.025±0.006 |
| Beat GPT-4o | NA | 5/8 | 7/8 | 5/8 |

*Table 5.* Performance Comparison of Different Token budgets for Gemini 2.5 Pro. We report mean MSE and standard deviation across datasets. Results that outperform GPT-4o are highlighted in green, while those performing worse are marked in red. In "Beat GPT-4o" row, we present the fraction of datasets where each reasoning budget outperforms GPT-4o.

*(a)* Results of Unimodal Short-term TSF Settings. We use numerical series only to forecast the next three months.

| Dataset | GPT-4o | Gemini 2.5 Pro 1K extended thinking | Gemini 2.5 Pro 2K extended thinking | Gemini 2.5 Pro 4k extended thinking |
|---|---|---|---|---|
| Agriculture | 0.021±0.011 | 0.916±1.270 | 2.712±0.000 | 2.712±0.000 |
| Climate | 1.599±0.500 | 2.099±0.000 | 2.099±0.000 | 1.951±0.210 |
| Economy | 0.631±0.135 | 0.439±0.204 | 0.583±0.000 | 0.557±0.037 |
| Energy | 0.363±0.110 | 1.239±0.835 | 1.829±0.000 | 1.288±0.764 |
| Flu | 0.568±0.425 | 3.171±0.000 | 3.171±0.000 | 3.171±0.000 |
| Security | 0.093±0.029 | 0.232±0.037 | 0.195±0.091 | 0.188±0.100 |
| Employment | 0.010±0.004 | 0.208±0.146 | 0.208±0.146 | 0.311±0.000 |
| Traffic | 0.385±0.471 | 0.141±0.085 | 0.201±0.000 | 0.162±0.056 |
| Beat GPT-4o | NA | 2/8 | 2/8 | 2/8 |

*(b)* Results of Multimodal Short-term TSF Settings. We use numerical series with textual context series to forecast the next three months.

| Dataset | GPT-4o | Gemini 2.5 Pro 1K extended thinking | Gemini 2.5 Pro 2K extended thinking | Gemini 2.5 Pro 4k extended thinking |
|---|---|---|---|---|
| Agriculture | 0.018±0.015 | 0.910±1.274 | 2.712±0.000 | 2.712±0.000 |
| Climate | 1.716±0.580 | 3.220±0.343 | 2.132±0.046 | 2.615±0.410 |
| Economy | 0.569±0.162 | 0.507±0.107 | 0.583±0.000 | 0.265±0.103 |
| Energy | 0.541±0.457 | 1.829±0.000 | 0.641±0.840 | 0.656±0.830 |
| Flu | 0.548±0.164 | 3.171±0.000 | 2.254±1.296 | 2.338±1.178 |
| Security | 0.076±0.052 | 0.200±0.083 | 0.192±0.095 | 0.187±0.102 |
| Employment | 0.020±0.006 | 0.311±0.000 | 0.105±0.146 | 0.210±0.143 |
| Traffic | 0.551±0.396 | 0.161±0.058 | 0.201±0.000 | 0.201±0.000 |
| Beat GPT-4o | NA | 2/8 | 1/8 | 2/8 |

*(c)* Results of Unimodal Long-term TSF Settings. We use numerical series only to forecast the next six months.

| Dataset | GPT-4o | Gemini 2.5 Pro 1K extended thinking | Gemini 2.5 Pro 2K extended thinking | Gemini 2.5 Pro 4k extended thinking |
|---|---|---|---|---|
| Agriculture | 0.093±0.057 | 0.856±1.179 | 2.523±0.000 | 0.859±1.177 |
| Climate | 0.754±0.051 | 1.277±0.000 | 1.277±0.000 | 1.604±0.463 |
| Economy | 0.463±0.146 | 0.359±0.000 | 0.359±0.000 | 0.359±0.000 |
| Energy | 0.197±0.038 | 1.396±0.000 | 1.396±0.000 | 0.968±0.604 |
| Flu | 0.219±0.053 | 1.920±0.000 | 1.920±0.000 | 1.920±0.000 |
| Security | 0.183±0.044 | 0.207±0.000 | 0.207±0.000 | 0.207±0.000 |
| Employment | 0.011±0.006 | 0.181±0.123 | 0.268±0.000 | 0.184±0.119 |
| Traffic | 0.066±0.046 | 0.414±0.000 | 0.284±0.184 | 0.284±0.184 |
| Beat GPT-4o | NA | 1/8 | 1/8 | 1/8 |

*(d)* Results of Multimodal Long-term TSF Settings. We use numerical series with textual context series to forecast the next six months.

| Dataset | GPT-4o | Gemini 2.5 Pro 1K extended thinking | Gemini 2.5 Pro 2K extended thinking | Gemini 2.5 Pro 4k extended thinking |
|---|---|---|---|---|
| Agriculture | 0.110±0.065 | 0.853±1.181 | 0.864±1.173 | 0.849±1.184 |
| Climate | 1.365±0.479 | 1.277±0.000 | 1.349±0.102 | 1.277±0.000 |
| Economy | 0.487±0.237 | 0.359±0.000 | 0.301±0.082 | 0.320±0.054 |
| Energy | 0.365±0.185 | 0.982±0.586 | 0.961±0.615 | 0.948±0.633 |
| Flu | 0.291±0.065 | 1.766±0.218 | 1.920±0.000 | 1.920±0.000 |
| Security | 0.196±0.056 | 0.207±0.000 | 0.144±0.045 | 0.207±0.000 |
| Employment | 0.015±0.002 | 0.094±0.123 | 0.268±0.000 | 0.182±0.121 |
| Traffic | 0.207±0.205 | 0.414±0.000 | 0.286±0.181 | 0.414±0.000 |
| Beat GPT-4o | NA | 2/8 | 3/8 | 2/8 |

*Table 6.* Performance Comparison of Different Token budgets for Claude-3.7-Sonnet. We report mean MSE and standard deviation across datasets. Results that outperform GPT-4o are highlighted in green, while those performing worse are marked in red. In "Beat GPT-4o" row, we present the fraction of datasets where each reasoning budget outperforms GPT-4o.

*(a)* Results of Multimodal Short-term TSF Settings. We use numerical series with textual context series to forecast the next three months.

| Dataset | GPT-4o | Claude-3.7-Sonnet 1K extended thinking | Claude-3.7-Sonnet 2K extended thinking | Claude-3.7-Sonnet 4K extended thinking |
|---|---|---|---|---|
| Agriculture | 0.018±0.015 | 0.009±0.005 | 0.006±0.001 | 0.008±0.004 |
| Climate | 1.716±0.580 | 1.304±0.265 | 1.376±0.032 | 1.437±0.082 |
| Economy | 0.569±0.162 | 0.145±0.102 | 0.167±0.043 | 0.308±0.044 |
| Energy | 0.541±0.457 | 0.131±0.047 | 0.101±0.045 | 0.171±0.071 |
| Flu | 0.548±0.164 | 0.545±0.217 | 0.473±0.258 | 0.777±0.223 |
| Security | 0.076±0.052 | 0.103±0.022 | 0.061±0.039 | 0.039±0.009 |
| Employment | 0.020±0.006 | 0.011±0.002 | 0.010±0.006 | 0.014±0.004 |
| Traffic | 0.551±0.396 | 0.044±0.004 | 0.043±0.003 | 0.059±0.019 |
| Beat GPT-4o | NA | 7/8 | 8/8 | 7/8 |

*(b)* Results of Unimodal Long-term TSF Settings. We use numerical series only to forecast the next six months.

| Dataset | GPT-4o | Claude-3.7-Sonnet 1K extended thinking | Claude-3.7-Sonnet 2K extended thinking | Claude-3.7-Sonnet 4K extended thinking |
|---|---|---|---|---|
| Agriculture | 0.093±0.057 | 0.124±0.037 | 0.081±0.029 | 0.046±0.014 |
| Climate | 0.754±0.051 | 0.699±0.091 | 0.679±0.015 | 0.684±0.019 |
| Economy | 0.463±0.146 | 0.053±0.004 | 0.186±0.063 | 0.057±0.029 |
| Energy | 0.197±0.038 | 0.169±0.055 | 0.209±0.022 | 0.255±0.079 |
| Flu | 0.219±0.053 | 1.157±0.574 | 1.137±0.140 | 0.782±0.467 |
| Security | 0.183±0.044 | 0.133±0.053 | 0.094±0.008 | 0.089±0.004 |
| Employment | 0.011±0.006 | 0.014±0.003 | 0.010±0.002 | 0.012±0.002 |
| Traffic | 0.066±0.046 | 0.021±0.010 | 0.161±0.179 | 0.026±0.013 |
| Beat GPT-4o | NA | 5/8 | 5/8 | 5/8 |

*(c)* Results of Multimodal Long-term TSF Settings. We use numerical series with textual context series to forecast the next six months.

| Dataset | GPT-4o | Claude-3.7-Sonnet 1K extended thinking | Claude-3.7-Sonnet 2K extended thinking | Claude-3.7-Sonnet 4K extended thinking |
|---|---|---|---|---|
| Agriculture | 0.110±0.065 | 0.027±0.009 | 0.035±0.011 | 0.057±0.029 |
| Climate | 1.365±0.479 | 0.819±0.048 | 0.859±0.096 | 0.783±0.067 |
| Economy | 0.487±0.237 | 0.097±0.032 | 0.084±0.028 | 0.069±0.028 |
| Energy | 0.365±0.185 | 0.097±0.061 | 0.091±0.049 | 0.141±0.042 |
| Flu | 0.291±0.065 | 0.724±0.143 | 0.566±0.248 | 0.729±0.394 |
| Security | 0.196±0.056 | 0.117±0.007 | 0.091±0.013 | 0.089±0.016 |
| Employment | 0.015±0.002 | 0.012±0.001 | 0.007±0.002 | 0.008±0.001 |
| Traffic | 0.207±0.205 | 0.041±0.013 | 0.039±0.025 | 0.033±0.006 |
| Beat GPT-4o | NA | 7/8 | 7/8 | 7/8 |

## C. Limitations and Future Work

Since reasoning strategies for foundational time-series models have not yet been studied and are difficult to implement directly, we have to reuse foundational language models to explore effective TSF reasoning strategies. We envision that our benchmark and insights offer promising potential for future research on understanding and designing effective reasoning strategies for zero-shot TSF. First, our results show that only limited existing System 2 approaches benefit zero-shot TSF. This can be interpreted as a distinction in reasoning logic between time-series analysis or zero-shot TSF, and other types of reasoning that are inherently learned by most System 2 models. This finding encourages future studies to design reasoning strategies more tailored to time-series data and zero-shot TSF tasks. Second, in validating the effectiveness of reasoning in zero-shot TSF, our open-source toolkit—comprising both REC4TS and TIME-THINKING —provides guidelines for future research. Specifically, we observe a test-time scaling trend for zero-shot TSF by incorporating reasoning within REC4TS, paving the way for future work on large-scale foundation TSF models with reasoning capabilities. Additionally, we provide TIME-THINKING, the first TSF dataset with reasoning annotations, enabling deeper investigation and understanding of reasoning strategies in zero-shot TSF.

## D. Conclusion

In this work, we introduce REC4TS, the first comprehensive benchmark to evaluate the effectiveness of existing reasoning strategies in zero-shot TSF tasks. Through REC4TS 's evaluations, we address two key questions: whether (RQ1) reasoning benefits zero-shot TSF, and what (RQ2) reasoning strategy is most effective. Our findings demonstrate that zero-shot TSF can indeed benefit from reasoning, and moreover, further demonstrate that self-consistency reasoning provides the most significant advantage in zero-shot TSF.

## E. More Details of REC4TS

### E.1. Details of Datasets

We provide details of datasets in Table 7 and visualization in Figure 5.

### E.2. Details of Reasoning Strategies

E.2.1. IMPLEMENTATION OF REUSING LLMS FOR UNIMODAL FORECASTING

Prompts used for unimodal time-series forecasting are detailed in Figure 7

E.2.2. IMPLEMENTATION OF REUSING LLMS FOR MULTIMODAL FORECASTING

Prompts used for multimodal time-series forecasting are detailed in Figure 6

E.2.3. IMPLEMENTATION OF CHAIN-OF-THOUGHT REASONING

Prompts used for chain-of-thought reasoning are detailed in Figure 8

E.2.4. IMPLEMENTATION OF SELF-CONSISTENCY REASONING

We set the default number of sampled reasoning paths in self-consistency to 3, aggregating numeric predictions by the median. Prompts used for self-consistency reasoning are detailed in Figure 9

E.2.5. IMPLEMENTATION OF SELF-CORRECTION REASONING

We set the default number of iterations for self-correction to 2. Prompts used for self-correction reasoning are detailed in Figure 10

### E.3. Details of Models

E.3.1. OPENAI MODEL SERIES

We select GPT-4o and o1-mini as representatives of System 1 and System 2 models from OpenAI. Released on May 13, 2024, GPT-4o is an autoregressive multimodal model which integrates all inputs and outputs into a unified neural network, offering faster and more efficient performance than its predecessors.[1] In comparison, o1-mini is trained with large-scale reinforcement learning to reason, and it shows better reasoning performance over GPT-4o. [2]

E.3.2. GOOGLE MODEL SERIES

Taken from the latest Gemini models, Gemini 2.0 Flash and Gemini 2.0 Flash Thinking compose a pair of System 1 and 2 models. Based on the non-thinking version, Gemini 2.0 Flash Thinking is an experimental model designed to enhance reasoning capabilities by generating its own "thinking process" during responses. This self-articulation allows the model to break down complex tasks into smaller steps, leading to more robust and accurate reasoning outcomes. [3]

---

[1] https://openai.com/index/gpt-4o-system-card/
[2] https://openai.com/index/openai-o1-mini-advancing-cost-efficient-reasoning/
[3] https://ai.google.dev/gemini-api/docs/thinking

*Table 7.* Overview of dataset in REC4TS, covering key variables across eight domains with weekly, or monthly frequencies

| Domain | Target | Frequency | Timestamps | Timespan |
|---|---|---|---|---|
| Agriculture | Retail Broiler Composite | Monthly | 496 | 1983 - 2024/04 |
| Climate | Drought Level | Monthly | 496 | 1983 - 2024/04 |
| Economy | International Trade Balance | Monthly | 423 | 1989 - 2024/04 |
| Energy | Gasoline Prices | Weekly | 1479 | 1996 - 2024/04 |
| Health | Influenza Patients Proportion | Weekly | 1389 | 1997 - 2024/04 |
| Security | Disaster and Emergency Grants | Monthly | 297 | 1999 - 2024/04 |
| Employment | Unemployment Rate | Monthly | 900 | 1950 - 2024/04 |
| Traffic | Travel Volume | Monthly | 531 | 1980 - 2024/04 |

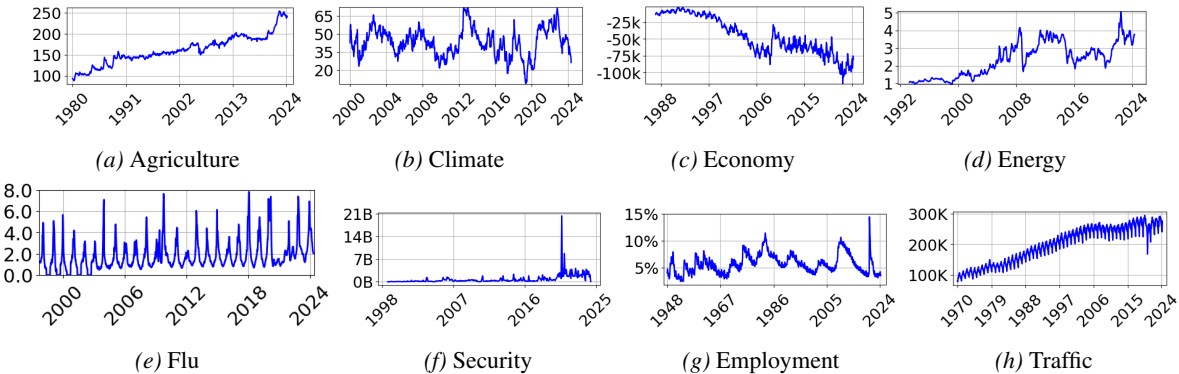

*(a)* Agriculture  *(b)* Climate  *(c)* Economy  *(d)* Energy

*(e)* Flu  *(f)* Security  *(g)* Employment  *(h)* Traffic

*Figure 5.* Visualization of time-series data.

### E.3.3. DEEPSEEK MODEL SERIES

DeepSeek-V3 is an open source 671B Mixture-of-Experts (MoE) language model. It employs Multi-head Latent Attention and DeepSeekMoE architectures for efficient inference and efficient training. DeepSeek-R1 surpasses V3 in reasoning tasks by incorporating cold-start data before reinforcement learning (RL), enhancing readability and performance. This approach distills reasoning patterns from larger models into smaller ones, resulting in superior reasoning capabilities compared to those developed through RL on smaller models. [4] [5]

### E.4. Details of Evaluations

We use the API from popular OpenRouter [6] platform to perform LLM inference. We strictly follow the recommended parameters for each LLM, i.e., median values from users on OpenRouter. We detail the LLM model versions and hyperparameter settings in Figure 11.

---

[4]https://github.com/deepseek-ai/DeepSeek-V3
[5]https://github.com/deepseek-ai/DeepSeek-R1
[6]https://openrouter.ai/

## F. Details of Test-Time Scaling Trend Experiments

### F.1. Data Setup

We adopted the TimesNet datasets as shown in Table 8 for our experiments in this section. All datasets listed are open source datasets.

### F.2. Model Setup

We utilized two popular zero-shot TSF models that support distributional forecasting, Chronos (Ansari et al., 2024) and Moirai (Woo et al., 2024), to carry out our investigation about relationship between self-consistency and model performance. Here we take the number of sampled paths as a reflection of self-consistency. We chose the "base" version in experiments for both models.

### F.3. Hyperparameters

The context for forecasting is set to 512 and the prediction length is set to 96, representing a short-term forecasting for 3 months. The specific numbers of sampled paths are set to 1 as a control group, then 5, 10 and doubling itself to 160 for scaling up.

```
base_prompt = f"""As an expert in {target_name}, predict the trends from {
    future_dates[0]} to {future_dates[-1]}
based on both historical data and contextual information.

Historical data (chronological order):
{historical_str}

Contextual information:
{context_text}

Please enclose your final predictions between [PRED_START] and [PRED_END]
    markers exactly like this:
[PRED_START]
2024-01-01: 123.45
2024-02-01: 124.56
[PRED_END]"""
```

*Figure 6.* Prompt used for multimodal time-series forecasting.

### F.4. Numerical Results

Tables 9–12 report original validation results for our test-time scaling trend, alongside the normalized results mentioned above.

## G. Details of Reasoning-Annotated TSF Datasets

We use GPT-4o with the agriculture and climate dataset as an example to demonstrate the reasoning annotated TSF dataset we built. We organize the data in a Question and Answering format.

### G.1. Agriculture

G.1.1. UNIMODAL TSF

Demo is provided in Figure 12.

G.1.2. MULTIMODAL TSF

Demo is provided in Figure 13.

### G.2. Climate

G.2.1. UNIMODAL TSF

Demo is provided in Figure 14.

G.2.2. MULTIMODAL TSF

Demo is provided in Figure 15.

*Table 8.* Dataset for Test-Time Scaling Trend Experiments

| Name | Domain | Frequency | Timestamps |
| --- | --- | --- | --- |
| Electricity | Energy | Hourly | 26305 |
| Traffic | Traffic | Hourly | 17545 |
| Exchange Rate | Economy | Daily | 7589 |
| ETTh1 | Energy | Hourly | 17421 |
| ETTh2 | Energy | Hourly | 17421 |
| ETTm1 | Energy | 15 Min | 69681 |
| ETTm2 | Energy | 15 Min | 69681 |

```
base_prompt = f"""As an expert in {target_name}, predict the trends from {
    future_dates[0]} to {future_dates[-1]} based on the historical data.

Historical data (chronological order):
{historical_str}

Please enclose your final predictions between [PRED_START] and [PRED_END]
    markers exactly like this:
[PRED_START]
2024-01-01: 123.45
2024-02-01: 124.56
[PRED_END]"""
```

*Figure 7.* Prompt used for unimodal time-series forecasting.

```
prompt = base_prompt + "\n\nLet's approach this step by step:\n1. Analyze
    historical trends\n2. Identify patterns\n3. Make predictions"
```

*Figure 8.* Prompt used for chain-of-thought reasoning.

*Table 9.* Numerical Results of Chronos (MSE)

| Num Path \ Dataset | Electricity | Traffic | Exchange Rate | ETTh1 | ETTh2 | ETTm1 | ETTm2 |
|---|---|---|---|---|---|---|---|
| 1 | 101164.19 | 2.17e-5 | 2.12e-3 | 13.95 | 50.00 | 9.62 | 37.42 |
| 5 | 63658.19 | 1.68e-5 | 1.14e-3 | 11.59 | 34.73 | 6.50 | 21.57 |
| 10 | 59627.12 | 1.60e-5 | 1.13e-3 | 10.29 | 31.32 | 5.94 | 20.71 |
| 20 | 55347.09 | 1.58e-5 | 1.02e-3 | 10.06 | 30.47 | 5.58 | 19.68 |
| 40 | 54779.91 | 1.55e-5 | 1.00e-3 | 9.98 | 29.96 | 5.60 | 18.86 |
| 80 | 54301.96 | 1.53e-5 | 1.00e-3 | 9.80 | 29.25 | 5.52 | 18.44 |
| 160 | 54233.27 | 1.53e-5 | 0.90e-3 | 9.98 | 29.52 | 5.50 | 18.54 |

```
elif method == "self_consistency":
    predictions = []
    for _ in range(3):
        pred = get_single_prediction(base_prompt, future, target_name,
            api_key, model)
        predictions.append(pred)
    return median_predictions(predictions)
```

*Figure 9.* Prompt used for self-consistency reasoning strategy.

```
elif method == "self_correction":
    current_pred = get_single_prediction(base_prompt, future, target_name,
        api_key, model)
    for _ in range(2):
        correction_prompt = f"{base_prompt}\n\nPrevious prediction:\n{
            format_prediction(current_pred)}\n\nPlease review and improve
            the prediction."
        current_pred = get_single_prediction(correction_prompt, future,
            target_name, api_key, model)
    return current_pred
```

*Figure 10.* Prompt used for self-correction reasoning strategy.

*Table 10.* Numerical Results of Chronos (MAE)

| Num Path \ Dataset | Electricity | Traffic | Exchange Rate | ETTh1 | ETTh2 | ETTm1 | ETTm2 |
|---|---|---|---|---|---|---|---|
| 1 | 225.51 | 2.77e-3 | 0.033 | 2.77 | 5.12 | 2.23 | 4.33 |
| 5 | 176.63 | 2.22e-3 | 0.024 | 2.51 | 4.31 | 1.83 | 3.40 |
| 10 | 168.59 | 2.17e-3 | 0.023 | 2.38 | 4.11 | 1.76 | 3.32 |
| 20 | 162.45 | 2.11e-3 | 0.022 | 2.35 | 4.05 | 1.69 | 3.20 |
| 40 | 161.03 | 2.07e-3 | 0.022 | 2.33 | 4.05 | 1.69 | 3.13 |
| 80 | 160.15 | 2.03e-3 | 0.022 | 2.29 | 3.97 | 1.67 | 3.11 |
| 160 | 159.89 | 2.03e-3 | 0.022 | 2.32 | 3.98 | 1.66 | 3.11 |

*Table 11.* Numerical Results of Moirai (MSE)

| Num Path \ Dataset | Electricity | Traffic | Exchange Rate | ETTh1 | ETTh2 | ETTm1 | ETTm2 |
|---|---|---|---|---|---|---|---|
| 1 | 181007.83 | 3.23e-5 | 3.80e-3 | 31.55 | 68.30 | 40.91 | 65.14 |
| 5 | 88376.75 | 1.73e-5 | 1.36e-3 | 14.57 | 42.92 | 8.45 | 32.44 |
| 10 | 79590.41 | 1.61e-5 | 1.23e-3 | 13.15 | 40.11 | 7.63 | 31.56 |
| 20 | 73638.04 | 1.57e-5 | 1.16e-3 | 12.31 | 37.42 | 6.91 | 27.54 |
| 40 | 71017.52 | 1.55e-5 | 1.11e-3 | 11.88 | 36.90 | 6.69 | 27.26 |
| 80 | 69676.72 | 1.53e-5 | 1.09e-3 | 11.70 | 36.39 | 6.58 | 27.23 |
| 160 | 69018.68 | 1.52e-5 | 1.08e-3 | 11.63 | 36.27 | 6.53 | 27.16 |

*Table 12.* Numerical Results of Moirai (MAE)

| Num Path \ Dataset | Electricity | Traffic | Exchange Rate | ETTh1 | ETTh2 | ETTm1 | ETTm2 |
|---|---|---|---|---|---|---|---|
| 1 | 277.88 | 3.21e-3 | 0.036 | 3.78 | 5.95 | 2.97 | 5.45 |
| 5 | 207.14 | 2.34e-3 | 0.026 | 2.77 | 4.84 | 2.09 | 4.25 |
| 10 | 196.03 | 2.23e-3 | 0.025 | 2.65 | 4.68 | 2.01 | 4.19 |
| 20 | 189.85 | 2.15e-3 | 0.024 | 2.52 | 4.50 | 1.88 | 3.87 |
| 40 | 185.28 | 2.11e-3 | 0.023 | 2.48 | 4.44 | 1.84 | 3.84 |
| 80 | 183.44 | 2.09e-3 | 0.023 | 2.46 | 4.44 | 1.83 | 3.84 |
| 160 | 182.38 | 2.07e-3 | 0.023 | 2.45 | 4.43 | 1.82 | 3.83 |

```
model_params = {
    "openai/o1-mini-2024-09-12": {
        "top_p": 1,"temperature": 0.7,"frequency_penalty": 0,"
            presence_penalty": 0,"repetition_penalty": 1,"top_k": 0
    },
    "google/gemini-2.0-flash-thinking-exp:free": {
        "top_p": 1,"temperature": 0.8,"repetition_penalty": 1
    },
    "deepseek/deepseek-r1": {
        "top_p": 1,
        "temperature": 0.7,
        "frequency_penalty": 0,
        "presence_penalty": 0,
        "repetition_penalty": 1,
        "top_k": 0
    },
    "openai/gpt-4o-2024-05-13": {
        "top_p": 1,"temperature": 0.9,"frequency_penalty": 0,"
            presence_penalty": 0,"repetition_penalty": 1,"top_k": 0
    },
    "google/gemini-2.0-flash-exp:free": {
        "top_p": 1,
        "temperature": 0.7,
        "frequency_penalty": 0,
        "presence_penalty": 0,
        "repetition_penalty": 1,
        "top_k": 0
    },
    "deepseek/deepseek-chat": {
        "top_p": 1,
        "temperature": 0.9,
        "frequency_penalty": 0,
        "presence_penalty": 0,
        "repetition_penalty": 1,
        "top_k": 0
    },
    "openai/o4-mini": {
        "temperature": "default API setting",
        "reasoning_effort": ["low", "medium", "high"]
    },
    "google/gemini-2.5-pro": {
        "temperature": "default API setting",
        "thinking_budget_tokens": [1024, 2048, 4096]
    },
    "anthropic/claude-3.7-sonnet": {
        "temperature": "default API setting",
        "thinking_budget_tokens": [1024, 2048, 4096]
    }
}
```

*Figure 11.* Hyperparameter settings for different models.

## Question ##
"As an expert in the US Retail Broiler Composite, predict the trends from
    2023−11−30 to 2024−01−31 based on the historical data.

Historical data (chronological order):
2015−11−30: 196.99; 2015−12−31: 193.15; 2016−01−31: 192.66; 2016−02−29:
    187.42; ...
2023−08−31: 252.59; 2023−09−30: 246.16; 2023−10−31: 247.64

Please enclose your final predictions between [PRED_START] and [PRED_END]
    markers exactly like this:
[PRED_START]
2024−01−01: 123.45
2024−02−01: 124.56
[PRED_END]"

## Answer ##
"To make a prediction for the US Retail Broiler Composite prices for
    November 30, 2023, to January 31, 2024, let's first analyze the
    historical trends from previous years in these months.

Here's a summary of the data for November, December, and January over the
    years:
− Nov 2015: 196.99
− Dec 2015: 193.15
− Jan 2016: 192.66
...
− Nov 2022: 252.10
− Dec 2022: 247.03
− Jan 2023: 242.33

There appears to be a pattern where November values are not substantially
    different from December, and changes into January can vary. Recent years
     indicate a trend of increasing prices into December followed by a
    modest decrease in January.

For projection, considering recent trends showing an upward movement in
    prices, we'll apply a similar rate of change to the most current data
    points.

Given that the last available price is for October 2023 at 247.64,
    predicting a reasonable increase in November, a slight upward or stable
    trend in December, and then a small drop or stabilization in January:

[PRED_START]
2023−11−30: 249.75
2023−12−31: 250.85
2024−01−31: 248.50
[PRED_END]"

*Figure 12.* Reasoning annotated example of unimodal agriculture.

## Question ##
"As an expert in the US Retail Broiler Composite, predict the trends from 2023−11−30 to 2024−01−31
based on both historical data and contextual information.

Historical data (chronological order):
2015−11−30: 196.99; 2015−12−31: 193.15; 2016−01−31: 192.66; ...
2023−09−30: 246.16; 2023−10−31: 247.64

Contextual information:
2021−12−30: The United States broiler market has seen a 2% decrease in broiler−type chicks placed, ...
2023−10−26: The annual broiler per−capita consumption is reported in the Today's Beef Consumer Fall 2023 Update.

Please enclose your final predictions between [PRED_START] and [PRED_END] markers exactly like this:
[PRED_START]
2024−01−01: 123.45
2024−02−01: 124.56
[PRED_END]"

##Answer##
"Based on the historical data and contextual information provided, here is my prediction for the US Retail Broiler Composite from 2023−11−30 to 2024−01−31:\n\n1. **Historical Data Analysis**:\n  − Historically, prices have exhibited seasonal patterns with a tendency to decrease towards the end of the year and then either stabilize or slightly increase early in the year.\n  − November to January has shown some degree of variability, with occasional drops in prices in December followed by modest increases or stability in January.\n\n2. **Contextual Information**:\n  − The US broiler market faces various influencing factors like international demand, supply chain disruptions, labor shortages, and prices of feed like corn.\n  − The ongoing issues such as price−fixing charges among major poultry producers and bird flu have potential impacts on both supply and prices.\n  − The US market is connected to global trends, including export demands which are showing strong signals for growth in 2024.\n\n3. **Market Trends**:\n  − From recent reports, there is an expectation of a higher supply of chicken meat but with weaker domestic demand, potentially leading to slight price reductions.\n  − However, the international market remains robust, and overall inflation pressures could balance out some of the downward trends.\n\n4. **Conclusion**:\n  − Given the current market conditions, we can expect a slight decrease in prices towards the end of 2023 compared to the middle of the year. However, prices might stabilize or slightly increase in January 2024 as the market adjusts to new supply−demand dynamics and international pressures.\n\n[PRED_START]\n2023−11−30: 243.80\n2023−12−31: 241.50\n2024−01−31: 243.10\n[PRED_END]"

*Figure 13.* Reasoning annotated example of multimodal agriculture.

## Question ##
"As an expert in the US Drought Level, predict the trends from 2023−11−30 to
    2024−01−31
based on the historical data.

Historical data (chronological order):
2015−11−30: 3.35; 2015−12−31: 4.05; 2016−01−31: 2.04; ...
2023−09−30: 2.12; 2023−10−31: 2.17

Please enclose your final predictions between [PRED_START] and [PRED_END]
    markers exactly like this:
[PRED_START]
2024−01−01: 123.45
2024−02−01: 124.56
[PRED_END]"

## Answer ##
"To predict the US Drought Level trends from November 30, 2023, to January
    31, 2024,
we need to analyze historical seasonal patterns and recent trends.

### Seasonal Trends and Analysis:
1. **Late Fall to Early Winter (Nov to Jan):**
    − Historically, November to January sees diverse behavior but often a
        decrease from late fall to early winter.
    − Novembers have shown values like 1.75, 1.62, 2.71, 1.94, 1.94, 1.30,
        2.44.
    − December and January values usually see some fluctuations, but
        generally, there's an inclination towards stabilizations or mild
        decreases.
...

3. **Data Synthesis:**
    − Given slight upticks as we move into winter and the fact that the
        October 2023 value is 2.17, we can predict a modest rise by end of
        November.
    − Typical increases from November to January stay within a moderate range
        , respecting the patterns seen in similar past periods.

### Predictions:
Considering the above factors, here are the expected values:

[PRED_START]
2023−11−30: 2.35
2023−12−31: 2.55
2024−01−31: 2.75
[PRED_END]

This prediction reflects historical patterns, recent trends, and typical
    seasonal behaviors."

*Figure 14.* Reasoning annotated example of unimodal TSF for US Drought Level.

```
## Question ##
"As an expert in the US Drought Level, predict the trends from 2023-11-30 to
    2024-01-31
based on both historical data and contextual information.

Historical data (chronological order):
2015-11-30: 3.35; 2015-12-31: 4.05; 2016-01-31: 2.04; ...
2023-09-30: 2.12; 2023-10-31: 2.17

Contextual information:
2021-12-30: Historical temperature, precipitation, and Palmer drought data
    from 1895 to present
for climate divisions, states, and regions in the contiguous U.S. are
    available.
2023-10-26: The contiguous U.S. had its 18th-warmest October in the 129-year
    record.

Please enclose your final predictions between [PRED_START] and [PRED_END]
    markers exactly like this:
[PRED_START]
2024-01-01: 123.45
2024-02-01: 124.56
[PRED_END]"

## Answer ##
"Based on the historical data and contextual information provided, I will
    make predictions for
the US Drought Level for the period from 2023-11-30 to 2024-01-31. I will
    take into account
the trends in recent years and the seasonal patterns observed in the
    historical data, as well
as the latest contextual information.

- Historical data suggests seasonal variations, with drought levels
    typically higher in the
  winter months.
- Recent contextual data indicate an expected wetter-than-normal winter for
    the southern
  contiguous U.S., which may alleviate drought conditions in those areas
      while potentially
  leading to more pronounced drought in others.

Considering these factors, here are the predicted drought levels:

[PRED_START]
2023-11-30: 2.25
2023-12-31: 2.55
2024-01-31: 2.35
[PRED_END]"
```

*Figure 15.* Reasoning annotated example of multimodal TSF for US Drought Level.

