# OpenReview forum: "Evaluating System 1 vs. 2 Reasoning Approaches for Zero-Shot Time Series Forecasting: A Benchmark and Insights"
_ICML.cc/2026/Workshop/FMSD — FMSD @ ICML 2026 Poster_

### Official Review · Reviewer_j2QP · 2026-05-16

**Rating:** 8
**Confidence:** 4

**Review:**

Summary of contributions

This paper presents REC4TS, a benchmark for evaluating reasoning strategies in zero-shot time series forecasting (TSF). The authors systematically compare four categories of reasoning approaches, including direct System 1 prediction, test-time enhanced System 1 methods (e.g., CoT, self-consistency, self-correction), System 2 reasoning models (e.g., DeepSeek-R1, o1-mini), and hybrid reasoning models with controllable reasoning effort. Experiments are conducted across multiple datasets and forecasting settings, including unimodal/multimodal and short-/long-term forecasting.

Strengths
1. The paper addresses an important and timely question: whether reasoning strategies developed for LLMs can benefit zero-shot TSF tasks. This direction is interesting given the growing popularity of foundation TSF models.
2. The benchmark is relatively comprehensive, covering multiple reasoning paradigms, model families, forecasting horizons, and multimodal settings. The large experimental coverage improves the empirical value of the work.
3. The observed scaling trend on probabilistic TSF models is insightful. Treating multiple sampled forecasts as reasoning paths and aggregating them via median prediction provides a simple yet effective form of test-time scaling for forecasting.

Weaknesses
1. Although the paper motivates reasoning as essential for zero-shot TSF, the actual performance improvements from reasoning are relatively limited in many settings. Most gains appear modest, and self-consistency is the only consistently effective strategy. However, self-consistency is fundamentally closer to ensemble averaging or variance reduction through multiple sampling paths rather than genuine multi-step reasoning. Therefore, it remains unclear whether the observed gains truly come from reasoning capabilities or simply from increased test-time compute and aggregation effects.
2. The comparison between System 2 reasoning models is heavily confounded by differences in model families, architectures, training data, and post-training pipelines. For example, the superior performance of DeepSeek-R1 cannot be conclusively attributed to GRPO-style reasoning training alone.
3. The claim that TIME-THINKING is “the first reasoning-annotated TSF dataset” appears overstated. There already exist several recent works exploring reasoning for time-series tasks, such as TimeOmni-1, TS-Reasoner, and related reasoning-enhanced TSF frameworks. The paper should position its contribution more carefully relative to these prior efforts.

Suggestions
1. The authors should include more detailed qualitative or mechanistic analyses of generated reasoning trajectories, such as examining whether reasoning traces correspond to interpretable temporal patterns, trend changes, uncertainty estimation, or event-aware forecasting behaviors.
2. The discussion of related work should be expanded, especially regarding recent reasoning-oriented TSF studies (e.g., TimeOmni-1, TS-Reasoner, and related reasoning datasets). The novelty claims around TIME-THINKING should be moderated accordingly.
3. It would also be interesting to investigate adaptive reasoning strategies, where the reasoning effort or number of sampled paths is dynamically allocated based on forecasting uncertainty or time-series complexity.

---

### Official Review · Reviewer_JLxp · 2026-05-19
**-**

**Rating:** 6
**Confidence:** 4

**Review:**

Summary:

The paper introduce REC4TS, a benchmark intended to evaluate various Large Language Model (LLM) inferences strategies (categorized into System 1 enhancements and System 2 models) applied to zero-shot time-series forecasting across eight domains. The benchmark tests foundation models in both unimodal and multimodal settings. The paper concludes that "self-consistency" is the most effective test-time strategy, while noting that "high" reasoning effort frequently degrades performance. The authors also contribute TIME-THINKING, a dataset of TSF samples paired with generated reasoning trajectories.

Strengths:
- Timely and Relevant Research Direction: Investigating how LLM inference paradigms map onto time-series forecasting is a highly relevant problem and and a good fit for the workshop
- Useful Baseline Artifacts: Running evaluations across multiple state-of-the-art models provides a useful empirical baseline for the community.
- Dataset Contribution: The release of the TIME-THINKING dataset is a practical contribution. Providing a curated set of reasoning trajectories over time-series data gives researchers raw material to experiment with distillation or fine-tuning approaches

Areas for Improvement:
- Semantic Overreach Regarding "Reasoning": The paper attempts to force standard statistical operations into an LLM reasoning taxonomy. Specifically, applying "Self-Consistency" to dedicated, probabilistic TSF models like Chronos and Moirai is a fundamental mischaracterization. Drawing multiple trajectories from a predictive distribution and calculating the median is standard Monte Carlo sampling, not a "reasoning path". Also the boundary established between "Test-Time Enhanced System 1" and "System 2" is porous. For instance, the authors classify Self-Correction as a System 1 enhancement. However, the process of generating an output, reflecting on its accuracy, and iteratively refining it heavily overlaps with standard definitions of System 2 analytical processing
- Unexplained Degradation in "High" Reasoning Effort: The paper observes (as shown in Figure 2) that extended reasoning tokens ("high" effort) frequently degrade performance compared to low or moderate effort, yet it is left largely unanalyzed. A qualitative error analysis would be intereseting.

Detailed Comments:
- I strongly suggest the removal of the experiment involving Chronos and Moirai under the guise of "Self-Consistency," or a complete reframing of this section. For probabilistic time-series models, drawing multiple trajectories from a predictive distribution and calculating the median is standard Monte Carlo sampling. It is not a "reasoning path" or "Self-Consistency. Rebranding standard sampling as "reasoning" undermines the credibility of the paper.
- I suggest to provide  qualitative error analysis of the outputs generated when the reasoning effort is set to "high." Understanding why the models fail when given more compute would be interesting

Justification:
This paper is an acceptable fit for a workshop venue because it tackles an relevant and important problem that might spark productive debate.

---

### Official Review · Reviewer_xSo3 · 2026-05-21

**Rating:** 6
**Confidence:** 4

**Review:**

> Summary

This paper proposes REC4TS, a benchmark for evaluating reasoning strategies in zero-shot time-series forecasting. It compares direct System 1 prediction, test-time enhanced System 1 methods such as Chain-of-Thought, self-consistency, and self-correction, post-training based System 2 models, and hybrid reasoning models with different reasoning-effort budgets. The benchmark covers eight domains, unimodal and multimodal settings, and short- and long-term forecasting horizons. The main findings are that self-consistency is the most reliable test-time reasoning strategy, low-to-moderate hybrid reasoning effort often gives the best trade-off, and multimodal TSF benefits more from reasoning than unimodal TSF. The paper also introduces REC4TS as an evaluation suite and TIME-THINKING as a reasoning-annotated TSF dataset.

Overall, the topic is timely and relevant to the ICML Workshop FMSD, since it studies foundation models for time-series forecasting and investigates whether LLM-style reasoning can improve structured predictive tasks. However, the current version has several methodological limitations that weaken the strength of its conclusions.

> Strengths

- Timely and relevant problem.
The paper studies whether reasoning strategies can improve zero-shot time-series forecasting, which is an important and underexplored direction for foundation models on structured data.

- Interesting multimodal forecasting setup.
The inclusion of aligned textual context alongside numerical time series is a meaningful extension beyond standard unimodal TSF and is potentially useful for real-world forecasting problems.

- Potentially useful released resources.
REC4TS, the test-time scaling analysis, and TIME-THINKING could be useful resources for future research if the benchmark protocol, dataset construction, and implementation details are made sufficiently reproducible.

> Weakness

- The comparison between LLMs and TSFMs is not well controlled.
The paper mainly evaluates LLM reasoning strategies on one set of time-series tasks, while the TSFM self-consistency experiment appears to use a different setup or dataset. This makes the comparison between LLM-based reasoning and TSFM-based test-time scaling difficult to interpret. If the authors want to argue that self-consistency or reasoning is broadly useful for zero-shot TSF, LLMs and TSFMs should be evaluated under the same datasets, horizons, metrics, and preprocessing protocol.

- The TSFM self-consistency setting is conceptually questionable.
The paper treats multiple sampled paths from probabilistic TSFMs such as Chronos and Moirai as "reasoning paths" and aggregates them by median. However, this is closer to probabilistic sampling or ensemble aggregation than reasoning. Therefore, the reported improvement may simply reflect variance reduction from multiple samples rather than evidence that a reasoning-inspired strategy benefits TSFMs. This part should be reframed as test-time ensembling or probabilistic path aggregation unless a clearer connection to reasoning is established.

- The research object and core insight are not clearly defined.
The paper adopts the System 1/System 2 terminology from outside computer science fields, but it does not provide a rigorous  definition for these concepts in zero-shot time-series forecasting. The evaluated factors include prompting strategies, sampling-based self-consistency, self-correction, different model families, post-training-based reasoning models, hybrid reasoning budgets, and TSFM sampling aggregation. These are heterogeneous mechanisms rather than clean variants of a single reasoning variable. Consequently, it is unclear whether the paper studies reasoning ability, test-time computation, prompting design, model-family effects, or ensemble-style forecast aggregation. This ambiguity also prevents the paper from delivering a clear core insight beyond several loosely connected empirical observations.

- Reasoning effects are confounded with sampling, ensembling, and model-family differences.
The strongest method is self-consistency, which samples multiple forecasts and aggregates them. This does not isolate reasoning from stochastic ensembling. Similarly, System 1 vs. System 2 comparisons are confounded because GPT-4o vs. o1-mini, Gemini-Flash vs. Gemini-thinking, and DeepSeek-V3 vs. DeepSeek-R1 differ in training data, model scale, alignment, inference behavior, and API implementation. The paper acknowledges this issue for DeepSeek-R1, but the main conclusions still risk attributing model-family effects to reasoning ability.

- The benchmark is incomplete for structured-data forecasting.
For a time-series forecasting paper, the current version lacks a robust evaluation protocal. It mainly reports MSE, while cross-domain TSF would benefit from normalized or scale-free metrics such as MASE, sMAPE, WAPE, or CRPS. In addition, the multimodal data construction process is not sufficiently clear. The authors claim to collect textual context from "keyword-based web summaries", but the paper does not provide enough detail on source selection, data cleaning, or leakage prevention. Since the paper's conclusion that multimodal TSF benefits more from reasoning depends heavily on the quality and temporal validity of the textual modality, this part requires much more careful documentation. Finally, the paper does not sufficiently report cost, latency, or token usage, which are particularly important for LLM-based structured-data construction.

> Minor Weakness

- Table formatting is inconsistent.
Several subtables in Tables 3–6 are not well aligned, which makes comparison across settings and model variants harder than necessary.

- Numerical precision is not standardized.
Tables 9–12 do not use a unified number of decimal places. This weakens readability and gives the results section a less polished appearance.

- Prompt examples are not clearly formatted.
Figures 6–15 contain prompt examples, but the layout is not sufficiently clear. It is difficult to distinguish system instructions, user inputs, model outputs, and forecasting targets.